# Neuropsychology and Electroencephalography in Rural Children at Neurodevelopmental Risk: A Scoping Review

Gilberto Galindo-Aldana *[ID] and Cynthia Torres-González [ID]

Laboratory of Neuroscience and Cognition, Mental Health, Profession, and Society Research Group, Autonomous University of Baja California, Hwy. 3, Col. Gutierrez, Mexicali 21725, Mexico; cynthia.torres.gonzalez@uabc.edu.mx
* Correspondence: gilberto.galindo.aldana@uabc.edu.mx

**Abstract:** Children from rural areas face numerous possibilities of neurodevelopmental conditions that may compromise their well-being and optimal development. Neuropsychology and electroencephalography (EEG) have shown strong agreement in detecting correlations between these two variables and suggest an association with specific environmental and social risk factors. The present scoping review aims to describe studies reporting associations between EEG features and cognitive impairment in children from rural or vulnerable environments and describe the main risk factors influencing EEG abnormalities in these children. The method for this purpose was based on a string-based review from PubMed, EBSCOhost, and Web of Science, following the Preferred Reporting Items for Systematic Reviews and Meta-Analysis (PRISMA). Qualitative and quantitative analyses were conducted from the outcomes that complied with the selected criteria. In total, 2280 records were identified; however, only 26 were eligible: 15 for qualitative and 11 for quantitative analysis. The findings highlight the significant literature on EEG and its relationship with cognitive impairment from studies in children with epilepsy and malnutrition. In general, there is evidence for the advantages of implementing EEG diagnosis and research techniques in children living under risk conditions. Specific associations between particular EEG features and cognitive impairment are described in the reviewed literature in children. Further research is needed to better describe and integrate the state of the art regarding EEG feature extraction.

**Keywords:** rural; well-being; infant welfare; risk factors

## 1. Introduction

Impaired cognitive development under risk conditions depends on different factors and how much they affect one another. Common causes compromising the cognitive and central nervous system (CNS) are often observed in disadvantaged social groups such as rural entities living in poverty [1]. The emergence of developmental cognitive neuroscience in the field of rurality research is both revolutionary and difficult; it represents difficulties stemming from both implicit and overt assumptions underlying this discipline. It requires a variety of rational theoretical alternatives for hypothetical psychological processes. The studies in this field might relate to particular brain functions and how various experimental events at various levels are connected. Studies reveal evidence of low socioeconomic status as a factor influencing neurocognitive differences between children from different socioeconomic backgrounds [2], including academic achievement and psychiatric outcomes but particularly language and executive functioning [3,4]. Neuroimaging studies have consistently evidenced patterns of association between socioeconomic status and brain development [5]; however, compared to other neuroimaging techniques, electroencephalography (EEG) is a viable clinical diagnostic and research technique. It is economically accessible, non-invasive, functionally sensitive, temporarily precise, and continuously improved for further complex feature extraction in different clinical conditions [6,7]. Classification methods for the sensitive detection of differences between normal and cognitively

impaired subjects through the analysis of the electrical activity in the brain have been explored and have shown good reliability, even using a small number of electrodes [8,9]. The EEG functionally probes the CNS to obtain a real-time record of electrical activity [10]. The origin of these electrical signals is in the pyramidal cells of the cerebral cortex; each neuron contains an electrical dipole that can be inhibitory or excitatory depending on the cell [11]. In this way, the method collects and records spectral information about the electrical activity from different brain regions through electrodes placed on the surface of the skull to capture the potential difference between them [12]. The standard EEG is a non-invasive, painless technique in which surface electrodes are attached to the scalp by a conductive gel and positioned according to the international 10–20 system. The voltage is measured between two electrodes. Usually, 16 to 24 electrodes are used for clinical purposes [13,14]; however, for research purposes, the number of electrodes can reach 128 electrode channels. Today, digital amplifiers are used, which facilitates signal analysis and storage as well as the ability to change parameters such as filters, sensitivity, recording time, and settings. A standard EEG, including activation techniques, primarily intermittent photostimulation and hyperventilation, should last at least 30 min [15]. These techniques are designed to provoke or enhance the appearance of abnormalities in brain activity. Neuron-related potentials recorded in the EEG are derived from the electrical activity of excitable tissues and gathered by measuring the potential difference between a scanner electrode and a reference electrode to investigate the biophysical basis of potentials of neuronal origin.

On the other hand, the conditions of rural development are part of society's history. They are not a new public health concern. However, most diagnoses related to brain maturation, specifically using EEG techniques, prevention, or treatment methods, are often developed and implemented in urban facilities, leading to challenges for rural living children [16,17]. Rurality in definition has wide variance; the Office Management and Budget (OMB) and other census institutions are often good resources for defining conditions for rural development. Accordingly, a rural area is defined as a nonmetropolitan area in which there is neither a city nor an urbanized area with 50,000 or more inhabitants, and the culture of the population who live in this condition is shaped by density, geography, agricultural heritage, economic conditions [18], social styles [19], religion, behavioral norms, and health care. Usually, there is a significant distance to health care services, different from urban communities [20], particularly neuropsychology and psychophysiology practice in rural settings [21], as well environmental factors that affect the nervous system like air pollution [22]. Rural areas represent disparities in health risk factors compared to urban residential assets. Rural residents usually demonstrate higher smoking rates and crude alcohol consumption [23]. The status of poverty or malnutrition is associated with behavioral conditions, such as depression in children [24]. A growing body of research shows that the fetal environment affects brain development and determines the brain's trajectory throughout the lifespan, particularly factors such as fetal alcohol exposure, teratogen exposure, and nutrient deficiencies [25].

These issues considerably influence and mold behavioral, mental health, and cognitive features. For example, the impact of mental health disorders and cognitive impairment in rural areas is higher compared to urban areas [26]. Children lack basic assets for a good quality of life, directly related to risk-taking behaviors and non-optimal cognitive development [27]. The literature suggests that neonatal mortality and morbidity are higher in rural and marginalized regions due to economic constraints, the absence of specialized obstetric and neonatal services, and the lack of awareness of the dangers to maternal and fetal health during pregnancy [28], which are linked to a higher prevalence of risk factors for early brain damage such as very preterm birth, low birth weight, hypoxic–ischemic encephalopathy, exposure to substances and maternal diseases, as well as septic processes [29]. In the case of the presence of early signs of neonatal seizures, EEG monitoring provides predictors related to unfavorable neurological signs, particularly in infants who experienced perinatal risks [30]. All these variables have been associated with an increased risk of neurocognitive developmental disorders.

Rural children, as compared with urban, show lower accuracy in visuospatial tasks measured by global and local visual features recognition and speed processing; they also present difficulties in internal image objects and visual memory [31]. According to this study, the lower abilities in visual feature processing relate to other neuropsychological domains such as letter processing, leading to the possibility of developing dyslexia, which according to Barbeiro and collaborators [32] is a widely underestimated neurodevelopmental disorder. A systematic review from Cainelli and collaborators [33] proposes that although learning disorders such as dyslexia have been widely studied, their brain electrophysiological causes remain not completely understood. In the report, they found most of the studies using spectral analysis of the EEG in resting states, including the specifics for activation or decrease in different brain lobes, as well as the non/significant differences. It is important to mention that children with dyslexia and children from control groups displayed abnormalities in a number of EEG measurements even when they were at rest, according to conclusions from the review, most notably an increase in delta and theta frequencies and a decrease in alpha frequencies, without any obvious localization. The same frequencies that were recorded while at rest seem to be connected to learning abilities. The current capacity to predict at-risk infants is still sub-optimal [34]; even indirect clinical familial risk of particular mood conditions, such as depression in mothers, is also considered a factor capable of modifying frontal alpha EEG activity in their children [35].

Several studies report the findings from the association of EEG abnormalities, patterns, or features. The scholarly literature addresses the relationship between congenital risk factors in children's development and EEG patterns, epilepsy, and brain development [36]. Previously integrated perspectives, such as the studies of the developmental origins of health and disease (DOHaD), strategically integrate a complex of studies, including observational epidemiological research from developmental exposures and its outcomes in later life as well as behavioral, psychiatric, and psychological dimensions [37]. Particularly in developing countries, the findings from the DOHaD paradigm address a framework for evaluating how early nutrition and growth affect long-term health. This collection of research demonstrates how early nutrition has a big impact on subsequent health and well-being [38]. In some rural areas, malnutrition developmental outcomes, overweight, and obesity are often neglected, representing a potential risk of future disease [39].

Previous studies associate EEG findings with cognitive impairment, and other literature reports the relevance of socioeconomic status and developmental conditions with children's health; however, few integrate and describe the main factors influencing children's brain development, including nutrition, exposure to neurotoxins, parasites, and other frequent risks found in rural assets. This scoping review integrates EEG data with neuropsychological (e.g., executive function, working memory, language, motor, and social skills), clinical, and behavioral (e.g., self-regulation, inhibitory control) conditions when children were exposed to neurodevelopmental risk factors. The aims of this scoping review are based on the following question: what are the main factors reported in the literature about the relation of EEG study-specific features and their relationship with cognitive impairment in children from rural areas or vulnerable environment factors? The present scoping review aims i. to qualitatively describe the significant findings from studies reporting associations between EEG features and cognitive impairment in children from rural or vulnerable environments and ii. to describe the main risk factors reporting an influence over EEG features in these children, focusing on the hypothesis that studies reporting data from children living in rural or vulnerable conditions present EEG abnormalities and cognitive impairment when exposed to neurodevelopmental risk factors in comparison to peers. To this end, we systematically reviewed studies reporting EEG findings and neuropsychological assessments from children living in rural areas or under vulnerable conditions.

## 2. Methodology

### 2.1. Search Strategy

The literature review was conducted following the Preferred Reporting Items for Systematic Reviews and Meta-Analysis (PRISMA) guidelines, identification, screening, eligibility, and inclusion for analysis [40] (Figure 1). The electronic databases for the search were PubMed, EBSCOHost, and Web of Science (WoS). Considering their disciplinary focus on the topic of interest, the selection of these databases is optimal for research in health sciences. We used a keyword string as follows: (("EEG") OR ("Electroencephalography") AND ("Children") OR ("Neurodevelopmental Risk"), AND ("Rural") OR ("Rurality") OR ("Vulnerable"), AND ("Cognitive Impairment")). Manual searching performed by G.G.A and C.T.G. of the reference lists of the obtained articles was also performed (the words used can be consulted in the supplementary materials). Outcomes from the manual search are described in the qualitative analysis section. For this review, we considered articles published through June 2023.

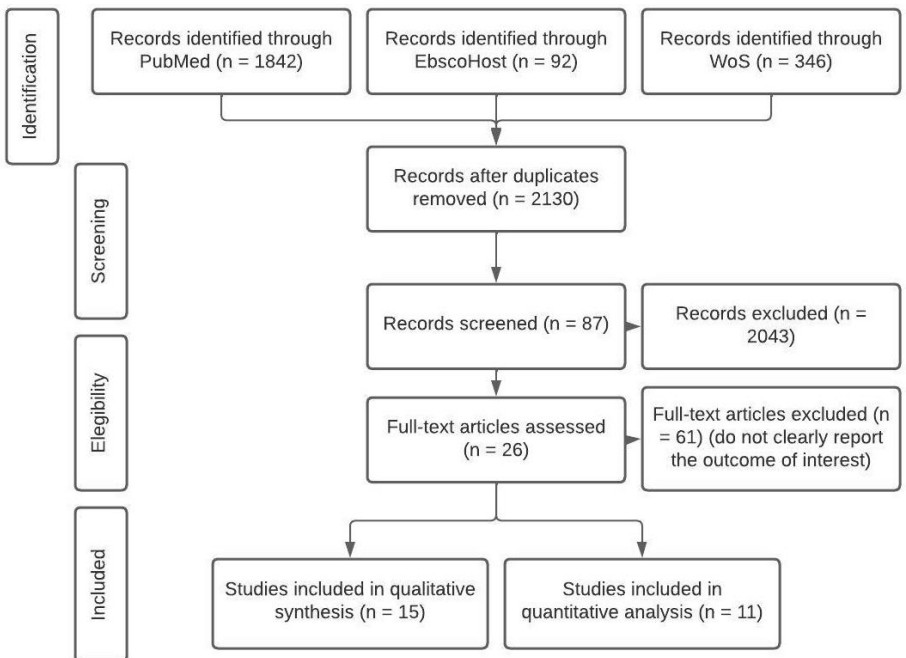

**Figure 1.** Search procedure flow chart.

### 2.2. Selection Process and Data Analysis

Outcomes from the search were divided into two groups. The first group of studies for qualitative analysis included research measuring visual EEG analysis, qEEG, and power spectrum comparisons, and their relationship with neuropsychological task performance and neurodevelopmental risk factors, covariates, country, and age of the sample. The description was subdivided from the relevance of the information reported, considering first EEG in children at neurodevelopmental risk and second EEG, environmental, and social risk factors. The data extracted from the literature were charted by continuous discussion by the two reviewers of this research, obtaining a final iteration of the results from the included reports. Quantitative analysis was only possible for studies reporting seizures and alpha and gamma power spectrum abnormalities, considered abnormal differences in children from rural or vulnerable conditions compared to a control group and also reporting these EEG features and their relationship with cognitive impairment. We used the reported *p*-value for group differences conclusions and the sample size to estimate Fisher's Z effect sizes, correlations, and confidence intervals. In both groups, the articles were included if they were published in peer-reviewed journals, were original, were written in English or Spanish, the participants were children from rural environments

or were raised under vulnerable conditions, and the use of any EEG recording of brain activity, visual or quantitative, power spectrum, and frequency domains, features, EEG connectivity analysis, or independent component analysis (ICA) was employed. Studies were also considered if reports were from data related to the resting state and presented an association with particular cognitive domains and risk factors or if they were during cognitive tasks. Systematic reviews, meta-analyses, and ERP studies, due to their specificity process, source-specific features, as well as time windows, or research not describing EEG features and their particular relationship with cognitive performance or impairment in children from rural or vulnerable environments were removed from the analysis.

**3. Results**

All the articles for the final inclusion described characteristics for research methods according to their aims. The participants' health-related conditions, study types, and study controls are shown in Table 1. Particular health-related factors influencing abnormalities in neuropsychological assessment and EEG are described in Table 2.

**Table 1.** Study types and controls and participants' health-related condition.

| Study Types (%) | Study Control (%) | Health-Related Condition (%) |
| --- | --- | --- |
| Prevalence (28) | Comparative (20) | Epilepsy-related events * (16) |
| Cross-sectional (16) | Randomly assigned (4) | Falciparum malaria (4) |
| Cohort (12) | Control groups (4) | Illiterate family (4) |
| Transversal Survey (12) | Assigned (4) | Preterm (4) |
| Case–control (8) | Randomized (4) | Studies reporting specific EEG health-related conditions (72) |
| Longitudinal (8) | Randomly (4) | |
| Age-matched (8) | Trial (4) | |
| Healthy controls (4) | Not clearly identified (56) | |
| Observational (4) | - | |

* Adverse perinatal events, head injury, and past history of febrile seizures.

*3.1. Qualitative Synthesis*

3.1.1. EEG in Children at Neurodevelopmental Risk

Living in rural areas is not a cognitive or central nervous system (CNS) development risk factor per se. However, this condition is frequently associated with exposure to pesticides affecting health [41], low-income [42], malnutrition in children and their parents [43], conditions which may directly or indirectly lead to cognitive impairment and electrophysiological abnormalities (Figure 2). Previous research seeking to describe an association between rurality and risk factors affecting mental health has shown that low schooling is the main related factor [44]. This problem is increased with the characteristics of rural communities that challenge the neuropsychological service delivery due to resource limitations, distance and costs, professional isolation, and beliefs about psychological services that reduce reliability all over these services [21].

Ueda and collaborators demonstrated a difference in frontal and occipital power variance function in the theta and alpha bands to be smaller in participants presenting mild cognitive impairment compared with healthy subjects [45]. Few studies have reported EEG features of suboptimal or impaired cognitive development in recent years. Rural or vulnerable environment development conditions characterized by low to middle income, farm-working as a sustainable productive activity, lack or deficient drain systems, treated water services at homes, nutrition deficiencies, and toxin exposure are linked to different particular neuropsychological and neurophysiological EEG features [46], which could be a predictive factor for anticipating risk or protective factors. However, these studies offer a

clear direction about the relevance of EEG features for analyzing cognitive development in children from at-risk rural environments. To analyze the effects of development risk factors and EEG techniques for their description, we selected a group of reports of systematic review on neuropsychological and neurophysiological studies in subjects with developmental risks with different neurophysiological characteristics associated with cognitive impairment. We summarize the studies in Table 2, which presents data of children aged 0 to 13 years who were exposed to risk factors from rural environments. Within the cognitive domain, research has highlighted language and IQ deficits as well as differences in motor and executive function skills between the study groups. These findings indicate a disadvantage in cognitive development for the at-risk study groups. Deviant EEG activity or cognitive skills can be detected through different sources of clinical analysis, which could be within a clinician's scope. It is strongly suggested that diagnostic and intervention practices performed by the neurophysiologist, neuropsychologist, or psychologist should pay special attention to possible predictive or alerting signals. For example, severe malnutrition is a common pattern in children from rural areas and has been shown to affect motor task performance, which is also evident in EEG patterns. In a series of case studies [46], EEGs of children from rural areas in undernourished conditions were visually analyzed, and it was found that these children had motor soft neurological signs related to specific abnormalities in frontal, sharp, and slow waves, becoming generalized, parietal sharp and slow waves and temporal slow waves.

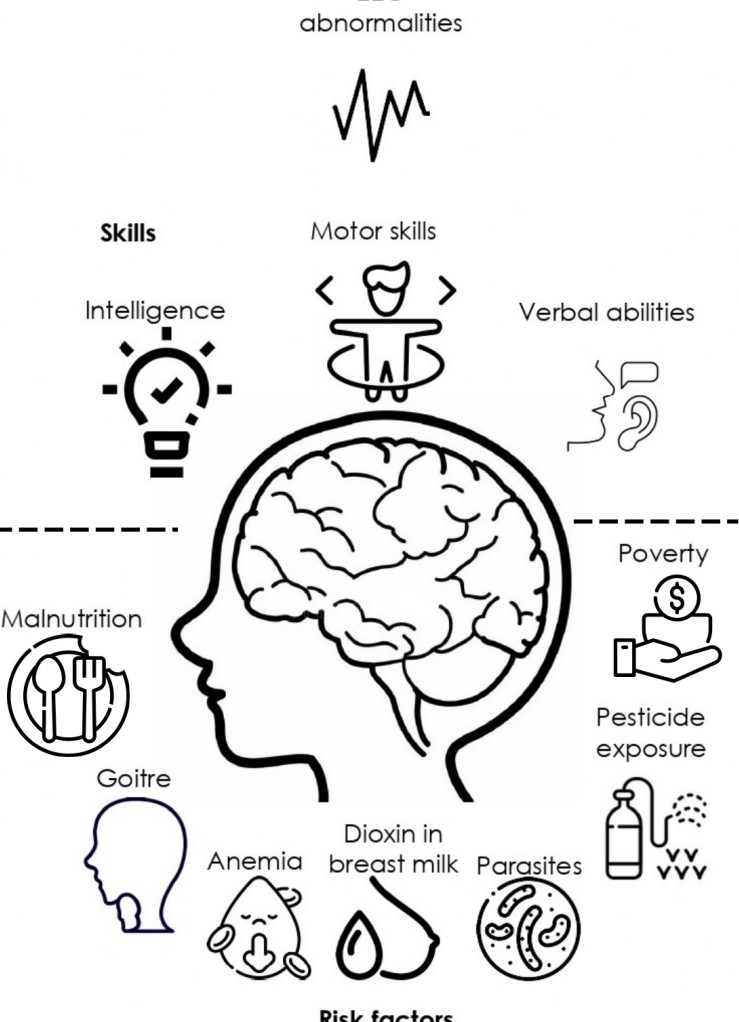

**Figure 2.** Main social, ecological, and health factors associated with EEG abnormalities and cognitive impairment, according to the literature analyzed for this study.

Later IQ scores can be predicted employing EEG pattern-measuring strategies even during early neonatal development periods, as suggested by previous studies. For example, in their study, Beckwith and Parmelee [47] proposed three visual methods to analyze EEG in preterm infants, one to describe unusual EEG patterns, a second to code the persistent remarkable delay on EEG maturation, and a third to describe the distribution of patterns within and across states of consciousness. Their results suggest that the integrity of the electrical neurophysiological organization of the brain has implications for later development, as measured in five-year follow-ups. Other severe EEG abnormalities such as epileptiform signs [48] and stress exposure have also been underestimated in at-risk populations. Several neuropsychological and EEG features have proven to have a high sensitivity to indicate a possible neurophysiological decline in conditions such as Parkinson's disease [49], autism [50], epilepsy [51], parental history of alcohol consumption [52], as well as cognitive impairment in areas such as memory, attention, language, and school success [53].

Thus, neuropsychological tests and EEG analysis are useful tools for studying cognitive impairment in at-risk populations. For example, qEEG analysis in Pakistani children suggested gamma power as a neural marker of cognitive function, specifically associated with executive function and verbal IQ [54]. Cognitive impairment in vulnerable rural populations is often severely underestimated or mistaken for other common developmental disorders. This is partly because most children with, for example, attention problems are impulsive and have low academic achievement. However, the cognitive and behavioral deficits associated with attention disorders are related to different brain impairments, such as less effective executive function, increased frontal theta waves, and a deficit in selective modulation of cortical activity [55]. Moreover, this underestimation is also due to the difficulty distinguishing between neurotypical development and mild cognitive impairment at the cognitive and electrophysiological levels. The diversity of cognitive and electrophysiological symptoms, according to the literature presented in Table 1, leads to the conclusion that there is not a general developmental disorder but a complex of non-optimal or mild multifactorial information processing conditions in children with attention or learning disorders or deviant behavior, which can be described by the characterization of EEG features.

In the same direction, the literature has widely reported that severe health risks are not the only condition causing abnormalities in children's bioelectrical functional state, with a significant association with other cognitive processes. In a study with 172 children, 10 to 12 years old, children with slight deficits in executive functions showed signs of suboptimal functional states of the limbic structures. Most evident behavioral deficits were related to motor and tactile perseverations and deviations in emotional–motivational regulation, such as poor motivation in task performance and poor communication skills, according to the authors [56]. In previous studies on children with learning difficulties, bilateral synchronous slow waves over the frontal and/or frontal and central cortices–frontal theta-waves seem to be the vector for the cortico–cortical functional connectivity for frontal regions and surrounding structures on the EEG, suggesting that frontal theta waves are most probably caused by the common frontal and central cortices' neuronal theta activity synchronized via cortico–subcortical links [57]. The findings consistently suggest that children with learning difficulties present functional connectivity abnormal patterns in theta, alpha, and beta frequency bands observed during resting state in the coherence of the functional coupling between the frontal and anterior temporal cortices found predominantly in the left hemisphere of frontal bilateral synchronous theta waves [58].

**Table 2.** Qualitative analysis of neuropsychological and electrophysiological features and their relationship with risk factors and covariates commonly found under rural development conditions.

| EEG Finding | Brain Region | Neuropsychological Finding | n | Covariates | Control Group | Age | Country | Associated with Factor | Ref. |
|---|---|---|---|---|---|---|---|---|---|
| Sharp slow waves, slow waves, generalized sharp and slow waves, sharp and slow waves | Right parietal, bilateral centroparietal, right frontal, bifrontal | Soft neurological signs, poor performance in motor tasks, successive finger tapping, heel–toe tapping, alternating hand pronation supination | 208 | Movement coordination disorders | 35.5% normally nourished | 8–10 year | India | Malnourishment | [46] |
| Lower gamma power | Frontal, and parietal | Better executive function performance, verbal intelligence | 105 | Anemia | Longitudinal study. No control group | From birth, 24, and 48 months | Pakistan | Poverty | [54] |
| Decrease in relative delta and increase in alpha and beta powers | Right frontal, and parietal | Positive correlation with language, and motor development | 55 | Gestational age, body length and head circumference | Longitudinal study. No control group | Prenatal 2-year follow-up | Vietnam (US) | Dioxin in breast milk | [59] |
| Lower relative alpha, and higher relative theta power | Bilateral central, temporal, and parietal | Delayed gratification and non-verbal cognitive ability. Lower scores in risk exposure group for visual reception | 143 | Friendliness | Non-adopted children group | 18 months | US (international adoption) | Adoption, deprivation, parental exposure to drugs, parental malnourishment and premature birth | [60] |
| Decrease in alpha, high theta | Lingual gyrus, and inferior frontal gyrus orbital right middle temporal gyrus | WISC full-scale IQ | 108 | Classification techniques | Healthy classmates matched by age, gender, and handedness | 5–11 years | Caribbean islands | Protein undernutrition | [61] |
| Centro-parietal slow-wave, paroxysmal, and focal abnormalities. Slow increment (<5 Hz). Decrease in alpha power (8.9 Hz) | Fronto-central. Centro-parietal, frontal | NA | 108 | NA | Control recordings | 5–11 years | Barbados | Protein undernutrition | [62] |
| Abnormal slow wave background EEG tracings, paroxysmal activity | Not specified | NA | 194 | Parasitism, and goitre, iodine level | Control group | 9–13 years | Ecuador | Malnutrition | [63] |

**Table 2.** *Cont.*

| EEG Finding | Brain Region | Neuropsychological Finding | n | Covariates | Control Group | Age | Country | Associated with Factor | Ref. |
|---|---|---|---|---|---|---|---|---|---|
| Bilateral slow waves, slow abnormal waves, sharp abnormal waves | Anterior brain areas, subcortical origin, Posterior regions | Reduced verbal abilities, problem solving/concentration, focus, and inhibition control/flexibility in at-risk groups | 194 | Infection protozoan parasite, parent's education | Control group | 9–13 years | Ecuador | Malnutrition | [64] |
| Alpha 1 band, and alpha-beta power ratio during driving 8 Hz | Temporo–occipital | NA | 20 | Lethargic movement, depressed oxygen consumption, and sodium pump activity | Healthy control group | 5–23 months | Jamaica | Malnutrition, marasmus and kwashiorkor | [65] |
| Synchronous theta waves | Frontal and limbic | Motor and tactile perseverations, emotional-motivational regulation, poor communication skills | 172 | Learning difficulties | No control group | 10–12 years | Russia | NA | [56,57] |
| Less beta and alpha power after stimulus repetition reduced U-shaped pattern | Central | Repetition and change detection responses predicting adaptive functioning at preschool age. | 63 | Intellectual and adaptive functioning | Normocephalic and macrocephalic children | 3–11 months and 2 years follow-up | Canada | Autism spectrum disorder, attention deficit disorder with hyperactivity | [66] |
| Reduced phase locking value in alpha, theta, delta, and beta | Prefrontal regions | Not specified | 128 | Demographic variables | Orphans and controls | 8–18 years | China | HIV/AIDS | [67] |
| Reduced total absolute band power (alpha, beta, delta, theta) | Fp1, Fp2, P3, P4, T3, T4, O1, and O2 | Neurocognitive outcomes and language skills | 48 | Intelligence and adaptive skills | Preterm infants and control <31 weeks of gestation | 12 h after birth and 2 years | Norway | Extremely premature birth | [68] |
| Abnormal EEG grading | Frontal, temporal, central, and occipital | Neurocognitive outcomes | 70 | Cognitive outcomes | Control group by neurocognitive and EEG results | 32–35 weeks 2 yr follow-up | Ireland | Preterm birth | [69] |

Note: NA, not applicable.

### 3.1.2. EEG, Environmental, and Social Risk Factors

Environmental exposure to toxins is present in rural and urban facilities, which leads to toxic concentrations of these compounds in different regions of the CNS. Furthermore, they may have a neurobiological effect even during prenatal in utero stages [70], leading to future developmental consequences. For example, exposure to lead and cadmium in children with learning disabilities in a rural population is related to differences in IQ performance, demonstrating that higher lead concentrations measured from hair are inversely correlated with intelligence and brain functioning [71]. In their study, Thatcher and Lester demonstrated, using a series of mixed methods of qEEG and EEG visual analysis that higher concentrations of lead or cadmium are associated with an increase in slow wave activity and a decrease in the amplitude observed in the EEG. The authors describe the different effects of exposure to heavy toxins on brain electrical activity. In particular, there is a strong correlation between the evoked potential measures obtained from the central lead and those obtained from the posterior occipital and parietal lead.

The low income in rural areas is, at the same time, a multiphasic factor that is allocated to different situations in adverse developmental settings. Children growing up under vulnerable conditions are frequently exposed to higher levels of stress due to stressful home environments explained by low income [72], anxiety [73], as well as prenatal cortisol exposure from the stressed mother [74]. Stress-related neurochemicals in early childhood, measured utilizing cortisol concentrations, are associated with impaired development of executive functions. In their study, Blair and Berry [75] describe findings that indicate children from 7 to 48 months with lower cortisol levels under resting conditions perform better on a battery of executive functions tasks. On the other hand, there there has been an advancement in the understanding of electrical neural activity assessed by using EEG. EEG power can be quantified among the different frequency bands and has been proven to be associated with different cognitive processes. Hence, studies suggest the analysis of power spectra distributions to understand better how neurons grow and become myelinated [76], which is closely related to cognitive development. External factors in children in vulnerable conditions represent a strong influence on brain development. For example, sleep deprivation associated with external stressors affects white matter myelin microstructure growth [77], and economic problems increase cerebrovascular diseases [78]. Neurological activation is dependent on cortisol concentrations during early infancy. In a study by St John [74], breastfeeding did not explain links between maternal cortisol and infant physiology. However, maternal cortisol at six months has proven to predict infant cortisol slope and EEG power even at 12 months, indexed by greater neural activation and reduced 6–9 Hz power, in a social interaction task. Furthermore, even in an advantaged, low-risk sample, infant neural activation, as indexed by 6–9 Hz power, is sensitive to subtle variations, especially maternal physiological regulation.

### 3.2. Quantitative Analysis

Quantitative analysis was taken from 11 studies reporting EEG abnormalities, mainly related to epileptic EEG features and seizure history. According to the data from children from rural areas, the reports have in common a first recognition based on epileptic history and a particular characterization of EEG features. Table 3 shows confidence intervals and standardized Fisher's correlation from sample size and *p*-values from the studies reporting an association of abnormal EEG and cognitive impairment. Six studies report a significant association between EEG abnormalities and cognitive impairment; the standard calculation from three studies showed a significant correlation between the two variables. A variety of poverty-related risk factors are commonly presented to infants and young children in low- and middle-income countries, increasing the possibility that these individuals may experience poor neurodevelopmental outcomes [79]. For the first instance, according to results from studies [80], inequalities in children's and teenagers' access to health care services and sociodemographic factors are considered as risk factors and are still present despite medical insurance coverage; for example, place of residence (rural/small city)

showed a $p < 0.01$. In particular, reports of the presence of malnutrition [81], acute encephalopathy, head injury before seizure onset, unfavorable prenatal events, and acute encephalopathy were the most likely proximate causes of convulsive features ($p < 0.001$) according to this analysis. Another significant association ($p < 0.001$) was found for data from children living in rural areas who presented co-morbidity of attention deficit with hyperactivity disorder and epilepsy [82]. Epilepsy and unfavorable prenatal occurrences have a high association according to research data [83]. Social and genetic variables may also be significant. Negative perinatal events had occurred in 14.3% of the 112 children from Burton's study but not in any of the controls, and 44% presented EEG abnormalities; taking into consideration the sample size of this study, we found a significant ($p < 0.001$) association of both factors. In data obtained from resting power and cognitive ability [54], better executive function was linked to increased EEG output in the gamma frequency bands (21–30 Hz and 31–45 Hz) in girls ($p < 0.036$). This may be taken into consideration as a protective factor in development.

**Table 3.** Quantitative analysis of EEG reports and cognitive impairment correlation.

| Reference | EEG Technique | n | Age | p-Value | r | 95% CI Upper Limit | 95% CI Lower Limit | Fisher's Zr |
|---|---|---|---|---|---|---|---|---|
| [80] | EEG seizures report | 1014 | 0–17 years | <0.01 | 0.0809 | 0.0194 | 0.1417 | 0.081 |
| [84] | ERPs | 50 | 6–7 years | 0.450 | - | - | - | - |
| [85] | ERPs | 178 | 4–12 years | 0.651 | - | - | - | - |
| [81] | EEG seizures report | 494 | † | <0.001 | 0.1476 | 0.0602 | 0.2328 | 0.1487 |
| [82] | EEG seizures report | 16 | † | <0.001 | 0.7419 | 0.3895 | 0.3895 | 0.9548 ** |
| [86] | EEG seizures report | 72 | 6–14 years | † | 0.3798 | 0.1624 | 0.562 | 0.3998 ** |
| [87] | EEG seizures report | 679 | † | † | 0.0833 | 0.0081 | 0.1575 | 0.0835 |
| [83] | EEG seizures report | 112 | 6–14 years | <0.001 | 0.126 | 0.0513 | 0.1994 | 0.1267 |
| [79] | ERPs | 148 | 1–5 months | 0.356 | 0.356 | 0.2065 | 0.4892 | 0.3723 ** |
| [54] | EEG gamma power | 105 | 0–24 months | 0.036 | 0.2049 | 0.0138 | 0.3816 | 0.2079 |
| [88] | EEG alpha and gamma power | 41 | 12–16 years | <0.01 | 0.0016 | −0.3062 | 0.3091 | 0.0016 |

† Data not directly reported from variables of interest. ** Significance $< 0.001$.

Access inequities to treatments in children who present neurodevelopmental risks are commonly reported; for example, sociodemographic research [80] showed that children with epilepsy aged 1 to 5 years old as opposed to other children and adolescents and children with epilepsy as opposed to children living in smaller cities and rural areas were more likely to receive specialized attention by neuropediatricians. Other countries, for example, reported that ERPs are effective for assessing children in rural areas [84,85] and are a useful tool to describe the neurophysiological maturity of the brain for the processing of visual novel information as well as face and auditory processing. Non-optimal development of sensorial brain systems such as brain somatomotor [89], or frontostriatal and cerebellar circuitry [90,91] can lead to clinical co-morbidity with ADHD. In the literature [82], this factor has suggested a complex clinical symptomatology that goes beyond that explained by the EEG features of status epilepticus. For example, in the research by Chidi and collaborators in Nigeria, the cross-sectional study in children with epilepsy, 14.2% of the sample had ADHD co-morbidity, being the most common inattentive subtype and significantly associated with poor academic achievement and living in rural areas.

According to a case–control study [86], co-morbidity in children with a history of epilepsy is very common among cases and is consistently associated with cognitive impairment (64%), behavior disorders (61%), and motor difficulties (26%).

## 4. Discussion

EEG is perhaps the most advantageous technique for research and diagnosis to implement in at-risk rural conditions. Neurophysiological factors underlie different cognitive impairment expressions, often underestimated in rural environments [39]. Many cognitive symptoms are often generalized and erroneously used in common clinical practice. The main findings of the qualitative analysis of the 15 papers included in this review showed

that early exposure to adverse conditions such as malnutrition, poverty, parental drug use, parental neglect, infectious diseases such as HIV/AIDS, and perinatal conditions such as prematurity are frequent risk factors associated with poor performance in motor coordination, language, executive functioning, and intelligence tasks, suggesting a negative impact on neurocognitive development and adaptive capacity in children with such conditions. The results of the behavioral tests were also linked to electrophysiological findings such as bilateral slowing; the presence of paroxysmal activity; low gamma power; decreased relative delta, alpha, and beta power; as well as increased theta power. On the one hand, the EEG reports address several mainly epileptiform features. However, slow bifrontal waves, a decrease in alpha and theta power, centro-parietal paroxysmal activity, abnormal background waves, and synchronous theta are commonly observed in later infants, while lower gamma power, a decrease in relative delta, and an increase in alpha and beta are reported from earlier developmental stages. Furthermore, on the other hand, basic neuropsychological assessments offer sensitive information about the cognitive performance of children and adults under rural risk living conditions.

This study presents evidence from at least two decades of EEG and cognitive impairment research, including the effects of neurodevelopmental risk factors among these variables. In addition, we can observe an advancement in EEG measurement techniques and feature extraction [92] with a reduction in cost and an increase in clinical sensitivity and classification of abnormalities. Research reveals that perhaps one of the most widely used techniques in rural areas is EEG feature description for seizure detection, being at a primary level accessible to a broad population, despite this being a clinically sensitive technique. According to the revised literature for this study, there is a significant relation suggested in at least 15 reports between EEG features and cognitive impairment in children with neurodevelopmental risk factors. These findings underscore a need for cultural indicators to inform neonatal, child, and adolescent assessment from birth onward, enabling early detection and, ultimately, intervention at critical neurodevelopmental stages. As protective factors, nutritional considerations such as blood iron biomarkers, behavioral measures of cognition, and EEG measurements of brain function have all been found to improve using a biofortified staple grain in dietary interventions [88]. The numerous processes by which blood iron content impacts brain function and cognition are strongly suggested by modeling the interactions among these variables.

Different conditions may lead children to live in left-behind conditions; such social factors could influence mental health among these children [93]. The recent literature suggests significant effects in neurodevelopment caused by endogenous as much as exogenous factors. For example, the exposome [94,95] is understood as a broad spectrum of factors affecting the genome in human beings. In addition, a robust body of recent research suggests that alterations in brain rhythms are central features of neurodevelopmental disorders, such as autism spectrum disorders. Longitudinal research has described significant changes in brain electrical activity, suggesting EEG may constitute a tool to establish biomarker changes during early neurodevelopment and represent a diagnostic measure with the potential to differentiate pathological conditions [96].

According to findings from this scoping review, clinical use of EEG is still improving in terms of behavioral and physiological appliances and scientific understanding. Long-term monitoring periods involving the use of digital analyses of particular cerebral measures can make use of effective and cutting-edge automated EEG and sleep state identification techniques that can also help with day-to-day clinical attention and the forecasting of neurodevelopmental outcomes [97] and neuroprotective strategies such as early maternal skin-to-skin contact [98].

Non-communicable diseases represent a risk factor for development in children, and early signs can be detected and reduced through social and dietary prevention actions [39]. In the framework of the DOHaD study of brain development, the early-life research must be expanded to include measures to prevent stunting and altered brain development, as well as to guarantee the long-term health and high quality of life of those who survive [99].

To this end, neuropsychological and EEG analysis and genetic and epigenetic exposome approaches [94] have demonstrated substantial contributions to understanding of children's development in recent years.

Further understanding of brain sources for specific skill development requires accuracy in brain EEG signal classification. Learning disorders, such as dyslexia, as a severely underestimated neurodevelopmental disorder [32], suggests an opportunity area for understanding children's brain electrical patterns in reading abilities [33]. Reading disorders in rural children are a crucial issue in neuroscience research closely related to other learning difficulties [31], and teaching [100,101] and parenting resources in rural areas may differ from those of urban settings [102]. Children with dyslexia manifest particular physiological EEG and heart rate variability parameters when they are resolving reading tasks, for example, longer saccade count, greater values of beta power, and broadband EEG (0.5–40 Hz) [103].

On the other hand, epidemiological data on epilepsy demonstrate a frequent interaction of this neurological disorder with cognitive impairment and abnormal patterns of brain development in children with myoclonic epilepsy [104]. The research on children who experience seizures reports impairment in domains such as executive functioning due to the detection of epileptic features in EEG; children frequently receive antiepileptic carbamazepine treatment, which, compared to other antiseizure medication, is shown to produce deterioration in cognitive functioning [105].

The role of parents in children's neurodevelopment is another crucial influencing factor; the findings from other research indicate that because left-behind children do not receive their parents' care for a prolonged period of time, their mental health requires attention and that its evaluation criteria are much greater than those of non-left-behind children. The most concerning aspect of left-behind children's mental health is that it differs greatly from other age groups regarding obsessive-compulsive disorder, interpersonal sensitivity, anxiety, aggression, paranoia, and mental disease. This is especially true for children aged 7 to 12 years old. It could be because children who are left behind between the ages of 7 and 12 are still developing, are less uninformed than those between the ages of 1–6, and are less mature than those between the ages of 13 and 17 [106].

Preterm birth is an excellent example of a condition with well-studied consequences on neurodevelopment at both early and long-term stages. This condition occurs due to multiple biological and environmental factors in the mother–placenta–fetus triad [107,108]. The literature suggests a strong relationship between factors such as brain development and preterm birth with very low weight [109], prenatal hypoxia in brain development, cognitive functions, and neurodegeneration [110]; other metabolic measures like insulin resistance measures are inversely associated with performance on cognitive tasks [111], while early hypoxic–ischemic insult is responsible for the underdevelopment of cortical gray matter and subcortical white matter at six months of age. It is capable of predicting poorer language development in early childhood [112], strongly supporting the need to increase efforts for raising awareness regarding cognitive impairment prognosis in children at risk.

Epidemiological studies show that most children with neurodevelopmental disorders live in countries in the Global South [108]. Infants who experience early social adversity are at increased risk for behavioral, emotional, and cognitive disturbances [113]. This vulnerability results from the complex interaction between endogenous and exogenous factors that occur more frequently in these countries and that include, as described in this work, genetic and epigenetic mechanisms and environmental stressors such as exposure to maternal infections, obstetric complications, exposure to natural or synthetic chemicals, exposure to environmental stressors, exposure to natural or synthetic chemicals and nutritional deficiencies during preconception and pregnancy, as well as demographic variables such as sex, socioeconomic status, parental education level, poverty, or living in a neighborhood with a high crime rate or with low access to green spaces free of environmental pollution [114–116].

Exposure to this variety of factors has multiple consequences that depend on the temporal window of exposure, resulting in a complex dimensionality of phenomena related to neurodevelopmental disorders. Considering correlations between simultaneous exposures, it is challenging to understand their effects [114].

The rate of neurodevelopmental disorders will continue to correspond with the decrease in infant mortality, which implies a challenge to ensure prosperity in childhood to ensure that children reach their developmental potential and to promote their integral well-being. This transition is in line with the Sustainable Development Goals of the United Nations, which refer to eradicating poverty and hunger, reducing maternal and infant mortality, and ensuring universal access to quality health services, inclusive and equitable education, and sustainable economic growth. Researchers and clinicians require reliable and valid tools to assess children in their specific contexts to achieve these goals [117].

Appropriate decision-making is crucial for children at high risk for neurologic sequelae. Early diagnosis improves prognosis and intervention outcomes. Thus, the use of accessible assessment and diagnostic methods for predicting developmental risk is necessary, especially in socially vulnerable populations [108,118].

## 5. Current Limitations and Future Directions

This scoping review has several limitations. The main effect of child abuse on neurodevelopment was not studied, thought this is another widely studied variable reported in the literature. Drug abuse and parental neglect are, for instance, significant risk factors commonly associated with neurodevelopmental clinical conditions and are frequently present in families living in rural areas in low-income socioeconomic communities. Another limitation of the study is the lack of information concerning protective factors; case–control studies, including the variables of interest in this analysis, are scarce. Further progress in the state of the art is required to achieve the possibility of a limited selection of electroencephalographic characteristics in the target population and its particular relationship with cognitive domains. There is a need to increase efforts to provide access to diagnostic services to children living in vulnerable or rural environments. It is also necessary to incorporate analysis related to EEG feature extraction, given the emerging developed techniques in this issue, which is promising for the identification of EEG abnormalities.

**Supplementary Materials:** The following supporting information can be downloaded at https://www.mdpi.com/article/10.3390/pediatric15040065/s1.

**Author Contributions:** Conceptualization, G.G.-A. and C.T.-G.; methodology, G.G.-A.; formal analysis, G.G.-A.; writing original draft preparation, G.G.-A.; writing review and editing, G.G.-A.; visualization, C.T.-G. All authors have read and agreed to the published version of the manuscript.

**Funding:** This research received no external funding.

**Institutional Review Board Statement:** Not applicable.

**Informed Consent Statement:** No informed consent was needed for this study.

**Data Availability Statement:** The data described in this study are available on request from the corresponding author.

**Conflicts of Interest:** The authors declare no conflicts of interest.

## Abbreviations

The following abbreviations are used in this manuscript:

EEG    Electroencephalography
qEEG   Quantitative Electroencephalography
CNS    Central Nervous System
IQ     Intellectual Quotient

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
