# Peer review of "Neuropsychology and Electroencephalography in Rural Children at Neurodevelopmental Risk: A Scoping Review"

_pediatrrep, doi:10.3390/pediatric15040065_

Round 1

Reviewer 1 Report

Comments and Suggestions for Authors

It was with great interest that I read the manuscript entitled: “Neuropsychology and EEG in Rural Children at Neurodevelopmental Risk: A Scoping Review” submitted to the Journal Pediatric Reports (Manuscript ID pediatrrep-2621311). In this study, the authors aimed to characterize associations between cognitive developmental trajectories reported in children from rural areas and (putative) neurophysiological electroencephalographic (EEG) indicators of vulnerability. A scoping review following the Preferred Reporting Items for Systematic Reviews and Meta-Analysis (PRISMA) was conducted to meet this goal. Despite the valuable contributions of this review, I would like to make some comments, suggestions, and questions, looking forward to have a reply from the authors with their comments and perspectives:

Title and abstract (page 1)

- This is just a minor comment: when reading the abstract, I missed more information or a note regarding the findings from the studies that were included in the review. The information provided seems too broad and does not inform the reader about some of the associations found between EEG and cognitive development indicators, for example.

Introduction (starting page 1)

The introduction presents relevant literature regarding the topic under analysis. Also, there is a clear statement of the main objectives. Nonetheless, I believe there are some aspects that would benefit from further clarification in order to showcase the motivations/arguments behind this review. Relatedly, the introduction would benefit of some reorganization so that the connection between ideas/arguments/evidence is smoother and more coherent. I leave here some examples:

- Page 1 (lines 22-25) - For me this sentence was bit convoluted. Perhaps, dividing it in smaller sentences while giving some examples could be helpful (e.g., where it reads “implicit and overt assumptions” or “various experimental levels of analysis might be connected”, some hints could be provided in parenthesis).

- Page 1 (lines 27-29) - Could the authors please give some examples that illustrate how the socioeconomic status is a relevant factor to consider when studying neurocognitive development? (some previous reviews might help; e.g., https://doi.org/10.1080/09297049.2021.1879766; https://doi.org/10.1038/s41380-023-02222-9; https://doi.org/10.1016/j.neubiorev.2021.08.027). This could also be an opportunity to introduce some neuroimaging data and establish the EEG as a suitable technique to collect data from populations for which it is not possible to obtain a behavioral response, such as infants and children.

- Page 1/2 (lines 29-54) - I have some small comments regarding the EEG:

i) Following the previous comment and more than describing general advantages of this technique, it would be paramount to show the perks of using EEG in early development and with vulnerable groups (otherwise one might wonder why is EEG special, why is it a good tool in clinical and experimental neurodevelopmental research, why did the authors focus only on EEG and disregard other techniques such as Functional Near-Infrared Spectroscopy – fNIRS, etc.);

ii) Could the authors please provide at least one citation that supports what is said in lines 33-36;

iii) The procedure described in lines 47-49 is not the only possibility to obtain an EEG recording, there are other approaches also used with children (e.g., https://doi.org/10.1016/j.neuroimage.2022.119508). Furthermore, if the authors are interested in particular EEG-related techniques, excluding, for example, event-related approaches, these options should to be addressed and justified.

iv) this is just a suggestion, but I was wondering if this detailed description of the technique would be more helpful after exploring a bit the main focus of the paper, particularly the characteristics and the challenges faced by rural communities.

- Page 3 (lines 126-139) - The authors mentioned the purposes of the study but drawing from the literature they have documented in the introduction, what are the main gaps that motivated the current scoping review(?), what sets this review apart from previous similar endeavors (exploring socioeconomic status to give an example) (?), ... This type of information might be helpful to establish a stronger connection between the evidence described in previous paragraphs and the study objectives.

Methods (starting page 3)

- page 4 (line 148) - The search terms used for the database search were the ones presented in the text or variations were also used? (e.g., children OR childhood OR infants OR infancy OR toddler*; “cognitive impairment” OR “cognitive development” OR cognit*). From what it is described, the keyword “neurodevelopmental risk” was used as an alternative of “children”, right? Could the authors please explain the rationale underlying this option, why not use it as a term related to “cognitive impairment”, for example.

Did the authors check references from previous systematic reviews (e.g., exploring the influence of socioeconomic status or childhood adversity)?

If the authors could provide the search expressions used in each database as a Supplementary Material, this could clarify the search without having to add more information to the manuscript. This would also aid future systematic reviews on this topic.

- page 4 (lines 149-150) - The authors refer that hand search was conducted from the reference lists of obtained records, but this is not reflected in the PRISMA flowchart. Does this mean that no record was identified via manual search? Regardless of the outcome, if this procedure was adopted, it should be depicted somehow in the search procedure.  

- page 4 (lines 159-161) - Did the authors compute any measure of interrater reliability (e.g., Cohen’s kappa)? What was the strategy adopted for disagreements?

- Did the authors extract any information regarding the quality of the records included in the review (especially considering the reports incorporated in the quantitative analysis)?

Results (starting page 5)

- Given the diversity of records included in the review, it would be interesting to have an initial subsection that offers an overview of the main research tendencies and methodological characteristics of the studies included in the review (e.g., participants’ sociodemographic and health-related characteristics beyond age; EEG-related procedures, analyses, indicators; cognitive outcomes and instruments used in the neuropsychological assessment).

Discussion (starting page 13)

The discussion raises important points and it brings together some key findings of the review in a brief and consistent manner. I leave here some small comments for the authors’ appreciation:

- As a reader, I missed a summary of the main findings. In other words, how can the authors reply to the question launched in the introduction: “(…) what are the main factors reported in the literature about the relation of EEG study-specific features and their relationship with cognitive impairment in children from rural areas or vulnerable environment factors?”. To illustrate, when the authors state: “According to the revised literature for this study, there is a significant relation suggested in at least 16 reports between EEG features and cognitive impairment in children with neurodevelopmental risk factors.” (page 13, line 370-377), what type of relationships were explored and reported? Are there putative EEG markers that might be relevant candidates to detect early signs of cognitive alterations in children and that merit further investigation? Given the wide range of ages included in the review (from infancy to adolescence), how can these different stages of development be considered in the overall discussion? (e.g., while some risk and protective factors are pervasive across development, others may change and have different weights).

These are just some examples and although I understand this might not be an easy task, some take home messages and practical implications are needed to aid the reader to navigate the multiplicity of results, which is something missing in the discussion in its current form. In a similar fashion and based on the evidence derived from this scoping review, a summary of the main gaps and limitations identified in the literature could be examined. This would be relevant to pinpoint avenues for future research that emerged as promising. This type of discussion will surely help researchers and professionals working in the field.

Minor comments:

- In general, the text seems well-written but there are some instances where an additional English revision might improve the reading fluency.

- Page 9-10 (Table 1) – Where it reads “Non” (e.g., third row, column “Associated with factor”), did the authors mean “None” or “Not applicable”?

- Regarding the reference list, there are some minimal lapses and inconsistencies (e.g., some titles are typed with the first letter in upper case, but some are written in lower case; as expected, some references have a “doi” but others do not include it).

Thank the authors for their contribution and best wishes for their work.

Sincerely,

Author Response

We appreciate reviewers’ suggestions and contributions to this manuscript. We took all of the comments in consideration, and made all the corresponding changes to the manuscript (all modifications now marked in blue).
Undoubtedly, all contributions helped substantially to improve the manuscript, we expect this new version may reduce the weaknesses of this manuscript

Reviewer 2 Report

Comments and Suggestions for Authors

The work presents an analysis of available literature on a very interesting and important topic. It is written clearly and coherently. It doesn't go too far with drawing conclusions, it just presents the status quo. What I miss in this text is the opposite view, i.e. what similar parameters look like in children living in cities, which also carry many factors impairing the development of the CNS (such as smog and other forms of environmental pollution, especially in industrial areas, small living space, stress resulting from living in crowded places, glaring differences in material status, etc.). Of course, it's hard - based on the above analysis - to draw far-reaching clinical or diagnostic conclusions - at least due to the paucity of literature on this topic, but it draws attention to a burning problem that, thanks to such publications, can be solved in a systemic way.

Author Response

The reviewer presents fundamental question related to basic approach of this proposal scoping review. On the table below, we address some considerations regarding to the opinions of the reviewer, and we point for new changes marked in blue in the new presentation of the manuscript.

Reviewer comment

Authors response

It doesn't go too far with drawing conclusions, it just presents the status quo.

We made an effort to objectively keep interpretations from the reviewed literature, presenting an integration of the main EEG findings related to cognitive impairment in children living in rural or vulnerable entities, and the main factors reporting influence on EEG parameters. We included new lines to address a synthesis of the main factors and challenges identified among the reviewed literature (Lines 374-388).

What I miss in this text is the opposite view, i.e. what similar parameters look like in children living in cities, which also carry many factors impairing the development of the CNS (such as smog and other forms of environmental pollution, especially in industrial areas, small living space, stress resulting from living in crowded places, glaring differences in material status, etc.). 

We consider the reviewer’s analysis very interesting, but at the same time very broad. It could be challenging to include in the same review the opposite side, this particular proposal refers to children who mainly live or were grown in rural environments, however, it does not discard the relevance of the effects of other risk factors present in urbinized entities. Future studies may compare these effects, but considering only comparative research.

Reviewer 3 Report

Comments and Suggestions for Authors

Dear Editor,

I want to express my thanks for giving me the chance to review such an interesting study. This review paper's major aim is to report the correlation between abnormal electrophysiological signals and cognitive abnormalities in children from disadvantaged and rural backgrounds. Additionally, the meta-analysis' methodology and findings are clearly laid out and detailed. In conclusion, I positively approve the manuscript in its current form.

Author Response

We thank the reviewer for the positive comments.

Round 2

Reviewer 1 Report

Comments and Suggestions for Authors

We would like to thank the authors for the effort and time devoted to address all the comments, suggestions, and questions. We acknowledge that efforts were made to address the concerns we raised. Overall, the changes resulted in an improved version of the manuscript. Accordingly, we are in favor of accepting the manuscript.

Thank the authors for their contribution and best wishes for their upcoming studies.

Sincerely,

Reviewer #1